# Leveraging Existing 16SrRNA Microbial Data to Define a Composite Biomarker for Autism Spectrum Disorder

YuShuang Xu,[a] YiHua Wang,[b] JinShuang Xu,[c] Yu Song,[d] BingQiang Liu,[b] ZhiFan Xiong[a]

[a]Division of Gastroenterology, Liyuan Hospital, Tongji Medical College, Huazhong University of Science and Technology, Wuhan, China
[b]School of Mathematics, Shandong University, Jinan, China
[c]Division of Nephrology, Jiaozhou Hospital of Tongji University DongFang Hospital, Jiaozhou, China
[d]Division of Gastroenterology, Jiaozhou Hospital of Tongji University DongFang Hospital, Jiaozhou, China

YuShuang Xu and YiHua Wang contributed equally to this work. The order was determined by the corresponding author after negotiation.

**ABSTRACT** Cumulative studies have utilized high-throughput sequencing of the 16SrRNA gene to characterize the composition and structure of the microbiota in autism spectrum disorder (ASD). However, they do not always obtain consistent results; thus, conducting cross-study comparisons is necessary. This study sought to analyze the alteration of fecal microbiota and the diagnostic capabilities of gut microbiota biomarkers in individuals with ASD using the existing 16SrRNA microbial data and explore heterogeneity among studies. The raw sequence and metadata from 10 studies, including 1,019 samples, were reanalyzed. Results showed no significant difference in alpha diversity of fecal microbiota between ASD and the control group. However, a significant difference in the composition structure of fecal microbiota was observed. Given the large differences in sample selection and technical differences, the separation of fecal microbiota between ASD and controls was not observed. Subgroup analysis was performed on the basis of different country of origin, hypervariable regions, and sequencing platforms, and the dominant genera in ASD and healthy control groups were determined by linear discriminant analysis (LDA) of the effect size (LEfSe) algorithm and Wilcoxon rank-sum test. Machine learning analyses were carried out to determine the diagnostic capabilities of potential microbial biomarkers. A total of 12 genera were identified to distinguish ASD from control, and the AUC of the training set and verification set was 0.757 and 0.761, respectively. Despite cohort heterogeneity, gut microbial dysbiosis of ASD has been proven to be a widespread phenomenon. Therefore, fecal microbial markers are of great significance in diagnosing ASD diseases and possible candidates for further mechanistic study of the role of intestinal microbiota in ASD.

**IMPORTANCE** This study provides an updated analysis to characterize the gut microbiota in ASD using 16SrRNA gene high-throughput sequencing data from 10 publicly available studies. Our analysis suggests an association between the fecal microbiota and ASD. Sample selection and technical differences between studies may interfere with the species composition analysis of the ASD group and control group. By summarizing the results of 16SrRNA gene sequencing from multiple fecal samples, we can provide evidence to support the use of microbial biomarkers to diagnose the occurrence of ASD. Our study provides a new perspective for further revealing the correlation between gut microbiota and ASD from the perspective of 16SrRNA sequencing in larger samples.

**KEYWORDS** autism spectrum disorder, fecal microbiota, 16SrRNA, biomarker

Address correspondence to ZhiFan Xiong, xiongzhifan@126.com, or BingQiang Liu, bingqiang@sdu.edu.cn.

The authors declare no conflict of interest.

Autism spectrum disorder (ASD) is a group of neurodevelopmental disorders characterized by deficits in social interactions, communication, and restricted and repetitive behavior (1, 2). Patients with ASD have a poor coping ability (3), low-quality of life (4), high rates of psychiatric comorbidity (5), and increased risk of suicidal behavior,

which brings heavy psychological and economic burdens to families and society. The World Health Organization (WHO) reported that one in 160 children worldwide has ASD, and the prevalence rate shows an increasing trend (6, 7). At present, the clinical diagnosis of ASD primarily based on the information gathered from a detailed history, physical examination, and the observation of specific characteristic behaviors, thereby limiting the remaining diagnostic biomarkers (7). The uncertainty of the potential etiology and unique pathogenesis of ASD and the sporadic effectiveness of existing treatment methods promote the exploration of specific diagnostic biomarkers and effective treatment strategies.

Gut microbiota plays a role in guiding and promoting brain development, and it has a long-term effect on health (8, 9). Gut microbiota and hosts cooperate to regulate immunity, metabolism, and nervous system development and function through the dynamic two-way communication of the "gut–brain axis" (10). Germ-free mice transplanted with gut microbiota from human donors with ASD exhibit hallmark autistic behavior, and treatment of ASD mouse models with candidate microbial metabolites can improve behavioral abnormalities and regulate neuronal excitability in the brain (11). In the ASD susceptibility gene Cntnap2–/– model for neurodevelopmental disorders, the researchers have found that different maladaptive behaviors are regulated by the interdependence of microbiota and host genes (12). These results indicate a potential link between the gut microbiota (or specific microbiota) and the brain in patients with ASD (10–12). The differences in species richness, diversity, composition, and structure of the gut microbiota between ASD patients and controls using 16S sequencing have been repeatedly reported, and they have received intense attention; however, a consensus among studies is rarely reported. For example, Zurita et al. found that *Bacteroides*, *Akkermansia*, *Coprococcus*, and different species of *Ruminococcus* increased in children with ASD relative to the controls (13). However, another study has found greater abundance of *Bacteroides* and *Prevotella* in ASD children and lower abundance of *Clostridium XlVa*, *Eisenbergiella*, *Escherichia/Shigella*, and *Akkermansia* (14). Disputes have also been considered to identify ASD-related microbial-based biomarkers. Large-scale multicenter studies using standardized methods may contribute to addressing these questions.

This study provided updated analysis to characterize the gut microbiota diversity in patients with ASD using 16SrRNA gene high-throughput sequencing data from 10 publicly available studies. Then, we evaluated the effect of study projects, country of origin, sequencing platform, and hypervariable region sequenced on the gut microbiota of patients with ASD. Finally, we identified different genera between the two groups using the LEfSe algorithm and Wilcoxon test and investigated whether these specific members of the community can be used as biomarkers to classify individuals as ASD or healthy controls accurately.

## RESULTS

**Grouping of ASD microbiota data sets.** The flowchart of study selection is shown in Fig. 1. The systematic searches obtained 792 records from the PubMed database and 10 projects from the GMrepo database. After preliminary screening of titles and abstracts, 51 records and one project were identified for full-text review. In addition, the 16SrRNA gene sequencing data from 10 microbiota studies met the criteria for further analysis. The 10 data sets were labeled as S1 to S10 (13–22). This combined data set consisted of 1,019 participants (569 ASD and 450 healthy controls), and the study sizes varied from 12 to 286 subjects. Two patients with ASD and four control samples in the S9 project were excluded because their number of valid tags was less than 6,000. The average age of two of the 10 studies was unclear, but all participants were minors. The sequencing platforms of all studies are Illumina. The sequencing fragments include V3 to 4, V4, and V4 to 5, and the research countries include China, Ecuador, Italy, and Korean. Detailed information on the data sets regarding demographic

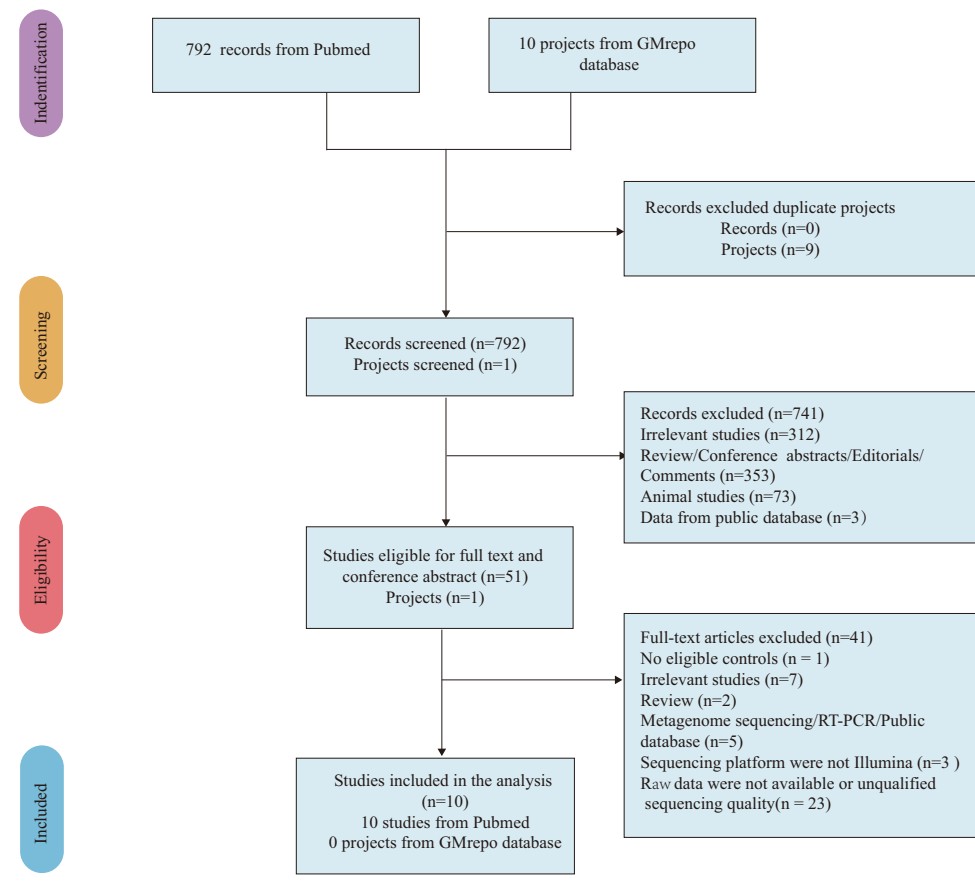

**FIG 1** Flowchart of the study selection.

characteristics, diagnostic methods of ASD, PCR primer, hypervariable regions, and informatics technology of studies included is depicted in Table 1.

**Alpha diversity analysis in patients with ASD and controls.** First, we assessed whether variation in alpha diversity was associated with the occurrence of disease. Analysis of fecal samples showed that no statistical differences were observed using the Shannon diversity index ($P = 0.782$), observed operational taxonomic units (OTUs) ($P = 0.356$), and Pielou's evenness index ($P = 0.822$), indicating that no significant difference in richness and uniformity of the gut microbiota was observed between ASD and control groups. Next, subgroup analysis was performed in accordance with different studies. In two of the 10 studies, the alpha diversity of the ASD group was lower than that of the healthy control group(all $P < 0.05$). The random model was selected for meta-analysis, and the results showed that no significant difference in Shannon diversity index (SMD = −0.113, 95% CI = −0.324 to 0.097, $P = 0.290$; $I^2 = 56.5\%$), observed OTUs (SMD = −0.120, 95% CI = −0.357 to 0.116, $P = 0.318$; $I^2 = 65.4\%$), and Pielou's evenness index (SMD = −0.231, 95% CI = −0.559 to 0.098, $P = 0.169$; $I^2 = 82.9\%$) was observed between ASD and control groups. The results are in Fig. 2 and Table S1. Further subgroup analysis was conducted on the basis of the country of origin, variable regions, and sequencing platform. and found that no significant difference in species evenness and richness was observed between ASD and healthy control groups. The results are shown in Tables S2 to S4.

**Cluster pattern of PCoA.** Ordination analysis based on Bray–Curtis dissimilarity (PERMANOVA, pseudo-F = 4.440, $P = 0.001$) and unweighted uniFrac distances (PERMANOVA, pseudo-F = 4.735, $P = 0.001$) revealed substantial variation among ASD and control groups regarding microbial community composition. The difference among groups was more significant that within groups (ANOSIM, $R = 0.023$, $P = 0.001$). However, the PCoA revealed apparent overlap distinguishing distributions between the two groups, in which the center points were close (Fig. 3a). Considering the significant differences in

**TABLE 1** Characteristics of included projects in the analysis

| ID | Study yr | ASD[a] n/age (mean) | Control n/ age (mean) | Diagnostic methods of ASD | PCR primer | Sequencing region | Country | Seq plat | Data availability (accession no.) |
|---|---|---|---|---|---|---|---|---|---|
| S1 | Chiappori et al. (19) 2022 | 6/NA | 6/NA | DSM-5 | 341F, 805R | V3 to 4 | Italy | Illumina MiSeq | SRA: PRJNA813424 |
| S2 | Chen et al. (20) 2021 | 138/6.1 | 60/6.7 | ADOS, CARS | NA | V3 to 4 | China | Illumina MiSeq | SRA: PRJNA769228 |
| S3 | Ha et al. (21) 2021 | 54/7.0 | 38/6.0 | DSM-5 | 341F, 805R | V3 to 4 | Korean | Illumina MiSeq | ENA: PRJEB45948 |
| S4 | Huang et al. (22) 2021 | 39/4.7 | 44/5.1 | DSM-5 | 515F, 926R | V4 to 5 | China | Illumina MiSeq | SRA: PRJNA687773 |
| S5 | Zou et al. (14) 2020 | 48/5.0 | 48/4.0 | DSM-4 | 338F, 806R | V3 to 4 | China | Illumina MiSeq | SRA: PRJNA578223 |
| S6 | Dan et al. (23) 2020 | 143/4.9 | 143/5.2 | DSM-5 | 515F, 806R | V4 | China | Illumina HiSeq 2500 | SRA: PRJNA453621 |
| S7 | Zurita et al. (13) 2020 | 25/8.9 | 31/8.5 | ADI-R | NA | V4 | Ecuador | Illumina MiSeq | SRA: PRJEB27306 |
| S8 | Zhao et al. (24) 2019 | 30/4.3 | 20/4.4 | ADOS, ADI-R | 338F, 806R | V3 to 4 | China | Illumina MiSeq | SRA: PRJNA624252 |
| S9 | Ding et al. (25) (2020) | 75/NA | 46/NA | DSM-5 | 515F, 806R | V4 | China | Illumina HiSeq 4000 | SRA: PRJNA589343 |
| S10 | Coretti et al. (26) 2018 | 11/2.9 | 14/2.9 | DSM-5, ADOS | NA | V3 to 4 | Italy | Illumina MiSeq | SRA: PRJEB29421 |

[a]ASD, autism spectrum disorder; DSM-5, diagnostic and statistical manual of mental disorders, 5th Edition; DSM-4, diagnostic and statistical manual of mental disorders, 4th Edition; ADOS, Autism Diagnostic Observation Schedule; ADI-R, Autism diagnostics interview-revised (ADI-R) questionnaire; CARS, Childhood Autism Rating Scale; Seq Plat, sequencing platform; SRA, Sequence Read Archive; ENA, European Nucleotide Archive; V1, V3, V4, V5, variable regions of the 16S rRNA gene.

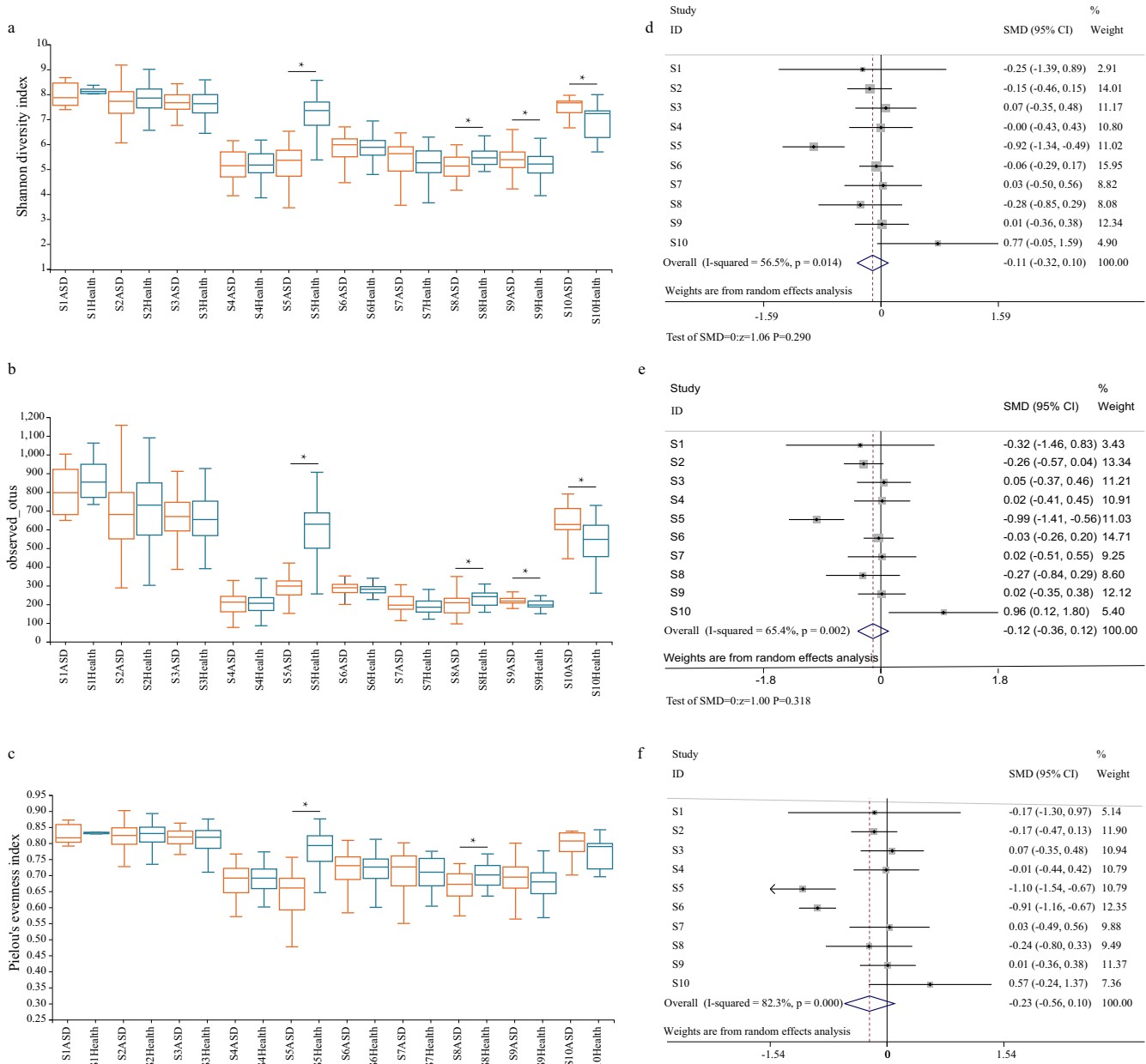

**FIG 2** Alpha diversity comparison between ASD and control groups using the Shannon diversity index (a), observed OTUs (b), and Pielou's evenness index (c) in different studies. The meta-analysis results of the alpha diversity in ASD and control groups. Forest plot displaying SMD and 95% CIs for the Shannon diversity index (d), observed OTUs (e), and Pielou's evenness index (f). *, $P < 0.05$ was characterized as significant difference.

some parameters, such as country of origin, variable regions, and sequencing platform, the batch effect in each study is too large. Therefore, further subgroup analysis was performed. The difference between the two groups based on different countries was less significant than that within groups (ANOSIM, $R = -0.032$, $P = 0.997$). However, the difference between groups based on different studies (ANOSIM, $R = 0.426$, $P = 0.001$), different variable regions (ANOSIM, $R = 0.178$, $P = 0.001$), and different sequencing platforms (ANOSIM, $R = 0.214$, $P = 0.001$) was more significant than that within groups.

The microbiota composition across ASD and healthy controls in subgroups was compared using PERMANOVA based on Bray–Curtis dissimilarities and unweighted uniFrac distance. Significant intergroup differences were observed in five out of 10 studies (S1, S5, S6, S9, and S10, all $P < 0.05$, Table S1). The gut microbiota composition between patients with ASD and healthy control sequenced using primers that targeted the V3 to 4 and V4 to 5

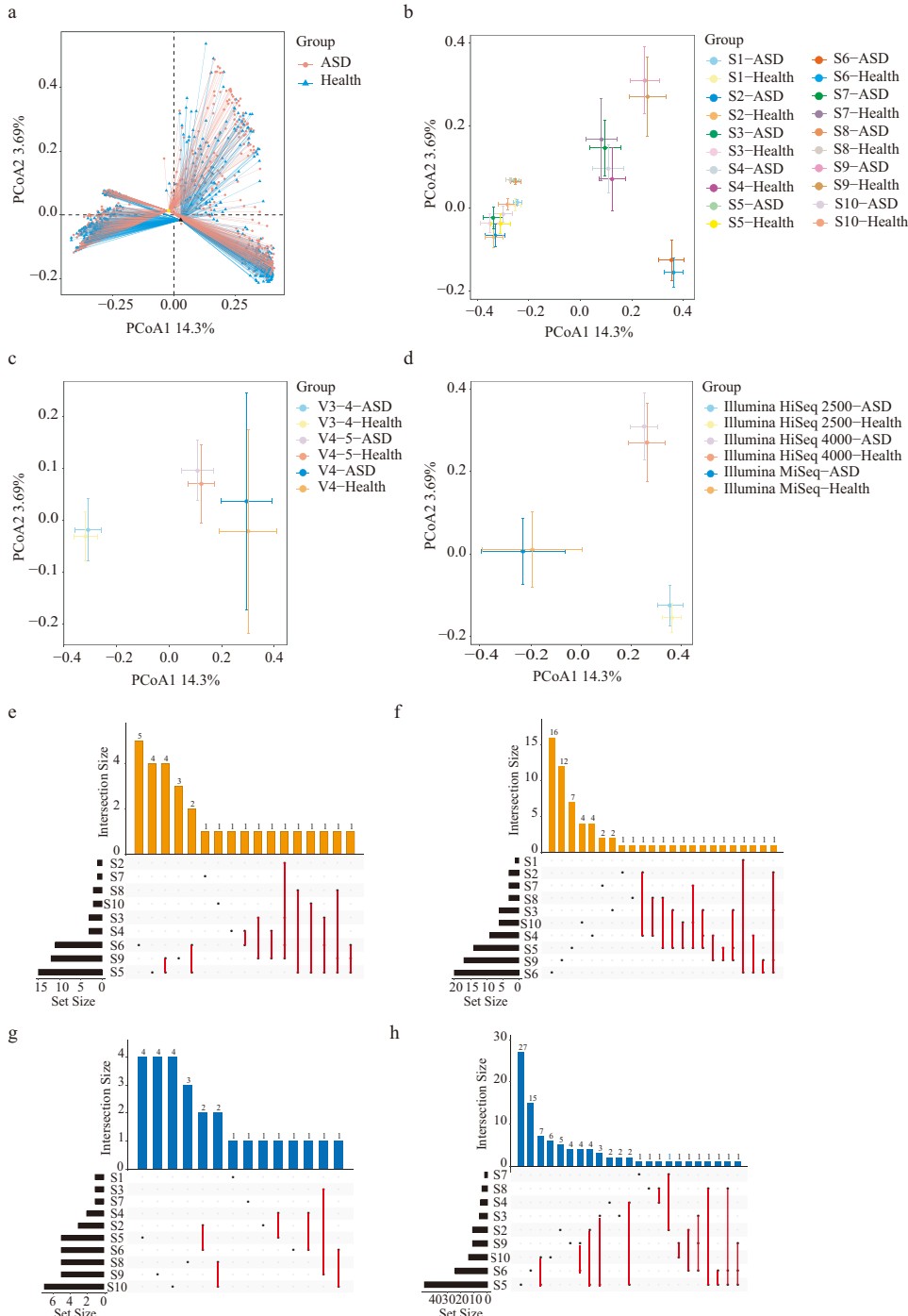

**FIG 3** Beta-diversity comparison between ASD and healthy control using Bray–Curtis dissimilarity (a). The PCoA based on Bray-Curtis dissimilarity shows the distribution of the microbiota composition structure of ASD and healthy control group grouped by different studies (b), variable regions (c), and sequencing platform (d). The overlapping feature of dominant genera in the ASD group between studies S1 to 10 using LEfSe algorithm (e) and Wilcoxon rank-sum test (f). The overlapping feature of dominant genera in the control group between studies S1 to 10 using LEfSe algorithm (g) and Wilcoxon rank-sum test (h).

regions differed significantly (all $P < 0.05$, Table S2). In addition, the gut microbiota composition between patients with ASD and healthy control sequenced using sequencing platform Illumina MiSeq, Illumina HiSeq 2500, and Illumina HiSeq 4000 differ significantly (all $P < 0.05$, Table S3). The PCoA plots based on Bray–Curtis dissimilarities between the ASD group and control group classified by projects, variable regions, and sequencing platform are shown in Fig. 3b to d.

**Identifying the fecal composite biomarker for ASD.** We identified the dominant genera in the ASD and control groups using LEfSe and Wilcoxon rank-sum test an included the intersection of the results in the two groups. The results showed that no significant dominant genera were found in the ASD group, whereas the dominant genera in the control group included *Ruminococcus 2*, *Oscillibacter*, and *Veillonella*. Next, subgroup analysis was performed on the basis of different studies, variable regions, and sequencing platforms. The dominant genera in ASD and control groups in studies S1 to 10 using LEfSe and Wilcoxon rank-sum test were screened (Tables S8 to S11). The overlapping feature of the dominant genera of ASD and control groups in different studies is shown in Fig. 3e to h. The dominant genera of the ASD group included *Lachnospiracea_incertae_sedis*, *Clostridium_XVIII*, *Eubacterium*, *Anaerostipes*, *Clostridium_sensu_stricto*, *Coprococcus*, *Dorea*, and *Faecalibacterium*. The dominant genera in the control group included *Gemmiger*, *Bacteroides*, *Roseburia*, *Dialister*, *Akkermansia*, *Haemophilus*, *Megamonas*, *Parabacteroides*, and *Streptococcus*. The results of dominant genera in ASD and control group subgrouped by three variable regions using LEfSe and Wilcoxon rank-sum test are summarized in Tables S12 to S15. The dominant genera of the ASD group included *Lachnospiracea_incertae_sedis*, and *Eubacterium*. The dominant bacteria in the control group included *Dialister*, *Prevotella*, *Megamonas*, *Parabacteroides*,*Clostridium_XVIII*, and *Roseburia*. Then, subgroup analysis was performed on the basis of different sequencing platforms, and the dominant genera in ASD and control groups were screened (Tables S15 to S19). The dominant genera of the ASD group included *Eubacterium*, *Bifidobacterium*, *Blautia*, *Dialister*, *Coprococcus*, and *Lachnospiracea_incertae_sedis*. The dominant bacteria in the control group included *Parabacteroides*, *Prevotella*, *Ruminococcus2*, *Romboutsia*, *Megamonas*, and *Clostridium_XlVa*.

Next, we used the dominant genera identified in different subgroups to build prediction models based on the random forest model. The top 12 important genera in the ASD and healthy control groups determined by subgroup analysis of different studies can distinguish the two groups, and the area under the receiver operating characteristic curve (AUC) of the training set and verification set was 0.688 and 0.706, respectively (Fig. 4a). The eight genera in the ASD and healthy control groups determined by subgroup analysis of different variable regions also can distinguish the two groups, and the AUC of the training set and verification set was 0.725 and 0.658, respectively (Fig. 4b). In contrast, the 12 genera in the ASD and healthy control groups determined by subgroup analysis of different sequencing platforms could distinguish ASD from control, and the AUC of the training set and verification set was 0.757 and 0.761, respectively (Fig. 4c).

## DISCUSSION

In this study, we found significant differences in fecal microbiota composition between ASD and healthy controls. However, the batch effect between the two groups was too large. Study projects, sequencing platform, and hypervariable region sequenced were important interference factors for intergroup differences among fecal sample groups, and the influence of the country of origin was relatively small. Then, we selected 12 dominant genera in ASD and healthy control groups by LEfSe and Wilcoxon rank-sum test, and these genera were analyzed to identify and validate microbiota-based biomarkers that could be used to classify individuals as ASD or healthy controls. Random forest classification models constructed to differentiate individuals with ASD from healthy controls using fecal samples showed good performance.

Compared with a single study, the data set of multiple studies can comprehensively detect the changes in ASD-related microbiota composition by increasing the sample size and evaluating interference factors. In our results, PCoA showed that the ASD group was closer to the control group in the same research project, indicating that the factors such as sample selection and technical differences greatly affected microbiota analysis. Previous studies have shown that geographical and ethnic groups are crucial in forming specific microbial communities (23). Chen et al. (24) found that the amplification of different hypervariable regions of the bacterial 16SrRNA gene (V1 to V2, V3 to

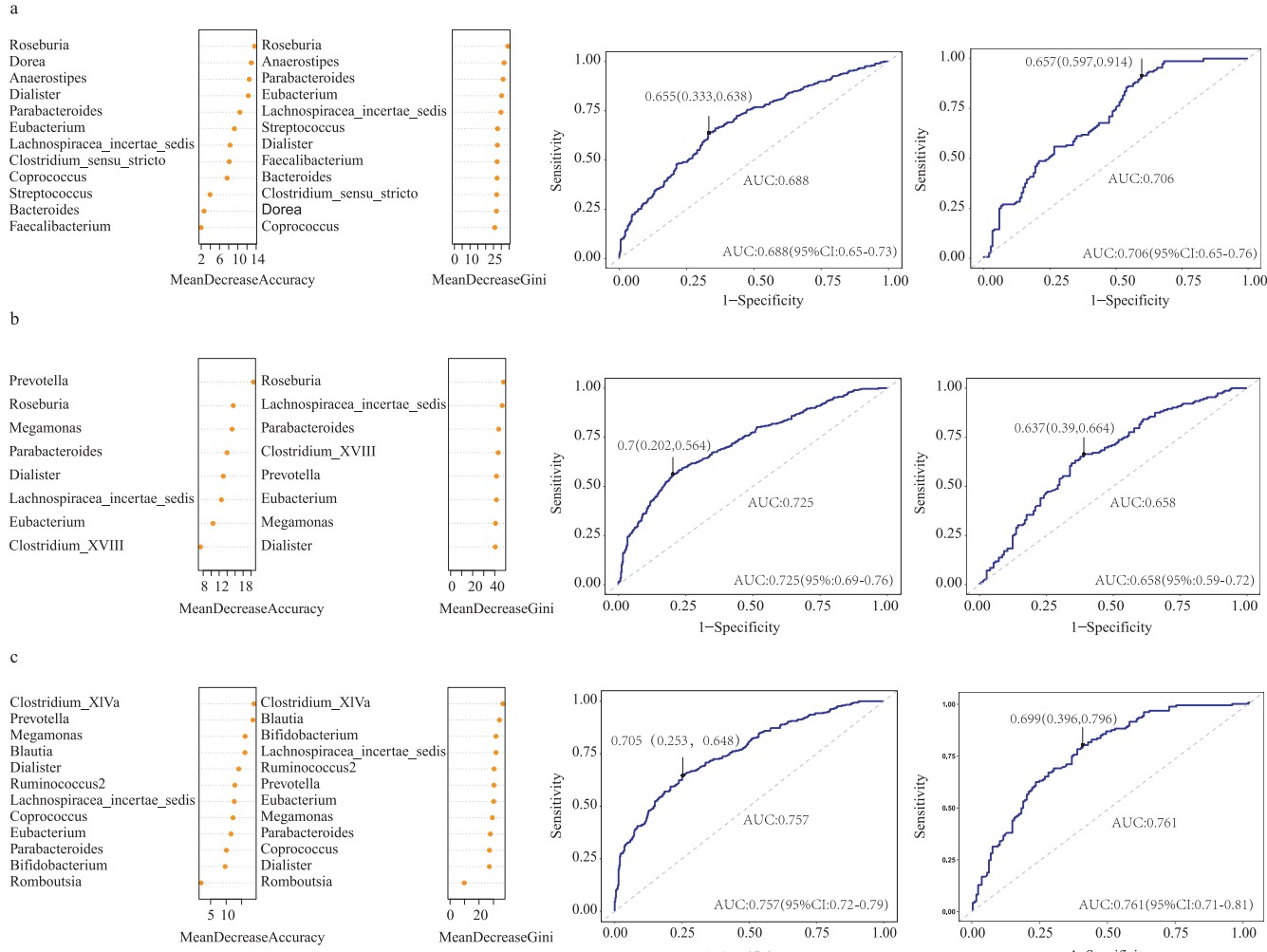

**FIG 4** RF model was used to build a predictive model of genus-level abundant genera. The relative importance of each genus in the predictive model was evaluated using the mean decreasing accuracy and Gini coefficient. ROC curve generated using genera determined by subgroup analysis of different studies (a), variable regions (b) and sequencing platforms (c).

V4, and V4) affected the identification of the entire microbial community and some bacterial groups. This result is consistent with our results, this is, the difference between groups based on different variable region targeted was more significant than that within groups. Batch bias caused by experimental protocols such as sample collection, sequencing platform, and bioinformatics analysis is important. Therefore, a consistent approach must be used to minimize batch deviations and facilitate comparison among data sets.

In a clinical treatment study, fecal bacteria transplantation can reduce the gastrointestinal and behavioral symptoms of autistic patients and change the composition and structure of intestinal flora, which is characterized by a significant reduction in the relative abundance of *Eubacterium coprostanogenes* (25). The offspring of LPS induced maternal immune activation showed an abnormal brain-gut-microbiota axis, accompanied by social behavior defects, anxiety-like and repetitive behavior, and ASD-like microbiota characteristics. The abundance of *Coprococcus* was relevant to the anxiety-like and repetitive behavior (26). The relative abundance of *Lachnospiracea incertae sedis* was significantly increased in children and adults (14, 27). In previous research, Sharon et al. (11) systematically analyzed the intestinal flora of ASD patients, ASD mice, and their offspring. They found significant differences among the gut microbiota of ASD patients, ASD mice, and the control group, and this difference can be vertically

transmitted to the offspring mice. *Verrucomicrobia*, *Proteobacteria phyla*, *Bacteroides ovatus*, and *Parabacteroides merdae* were significantly enriched in normal individuals/mice. Spearman correlation analysis showed that *Bacteroides* and *Parabactoids* were positively correlated with the decrease of repetitive behavior and increased social behavior. The use of microbiota transfer therapy can significantly improve the relative abundance of *Bifidobacterium*, *Prevotella*, and *Desulfovibrio* in patient with ASD, and these changes persisted even after the cessation of treatment. Other studies have shown that the ASD group has lower levels of *Romboutsia* and *Megamonas* (27, 28). These results partially overlaps with our results. Based on previous physiological studies, the biomarkers that we identified, likely play a pathogenic or therapeutic role in the occurrence of ASD. However, several studies have concluded that the relative abundance of *Bifidobacterium*, *Blautia*, and *Dialist* in patients with ASD is significantly reduced, which is different from our results (29). This is a controversial issue that needs further study.

Notably, we have not consistently identified bacteria in 10 intestinal microecology studies, that is, bacteria with an intersection. This result may be due to the relatively small number of individuals involved in limited studies. Their sporadic distribution among individuals shows that the role of the gut microbiota in ASD may be overestimated, but many mechanisms can lead to ASD, or a variety of bacteria can mediate a single mechanism (such as inflammation). For example, children with ASD often exhibit an immune response disorder. Using the mouse model of maternal immune activation, Kim et al. (9) found that the increase of IL-17A in maternal inflammation cannot only increase the risk of developing neurodevelopmental disorders but also affect the CD4+T cells of the offspring through the changes of gut microbiota, resulting in an immune sensitization phenotype of the offspring, thereby increasing susceptibility to bacteria-induced gut inflammation. In addition, the interaction between the gut microbiota and mammalian nervous system forms both adaptive and dysfunctional neurological processes through the "gut-brain axis" pathway. Moreover, intestinal inflammation caused by gut microbiota imbalance can cause intestinal barrier dysfunction and induce systemic inflammation, which may lead to structural changes in the brain barrier and induce mental symptoms (30, 31). Gut microbiota and its metabolites can also affect the brain and behavior by acting on the vagus nerve and intestinal nervous system (32). However, sufficient experimental evidence to support this finding is lacking.

**Strengths and limitations.** This study can identify some studies and pool their results, with a relatively large number of participants. It is also an update and extension of previous similar studies. Nevertheless, this study has several limitations. First, the metadata information of existing research is incomplete. The composition of individual feces varies greatly, which is affected by many confounding factors, such as region, diet, exercise, gender, age, body mass index, and specimen collection method. Therefore, we hope to integrate other possible confounding factors in future research. Second, the projects included in this study involve different sequencing platforms and variable regions targeted. We tried to eliminate the batch effect by using the combat function (33) and removeBatchEffect limma function (34), but it was not successful. Large batch effects can still be observed among different projects. Therefore, the method of eliminating the batch effect must be developed and utilized more. Third, whether intestinal disorders precede the development of ASD or whether restrictive and repetitive behaviors such as diet in ASD drive this disorder remain unclear. The dynamic changes in the relative abundance of individual features in time sequence may be used to identify the driving pathogenic factors in the development of ASD. Large-scale human microbiota research must collect samples from multiple time points to maximize the detection of minor effects in microbial host interactions.

**Conclusion.** Our analysis suggests an association between the fecal microbiota and ASD. Sample selection and technical differences among studies may interfere with species composition analysis of the ASD group and control group. By summarizing the results of 16SrRNA gene sequencing from multiple fecal samples, we can provide evidence to support the use of microbial biomarkers to diagnose the occurrence of ASD. Our study provides a new perspective to reveal the correlation between gut microbiota

and ASD based on 16SrRNA sequencing in large samples. Nevertheless, future research is necessary to understand the exact contribution of changing intestinal microbiota to the clinical manifestations of ASD, which lead to the development of prevention and treatment methods.

## MATERIALS AND METHODS

**Study search, selection, and inclusion.** We searched within the PubMed database. The search strategy and search terms were constructed as follows: (((((Autism Spectrum Disorder [MeSH Terms]) OR (Autism Spectrum Disorder [Title/Abstract])) OR (Autistic Spectrum Disorder [Title/Abstract])) OR (Autism [Title/Abstract])) OR (Autistic Disorder [Title/Abstract])) AND (((((((Microbiota [MeSH Terms]) OR (Microbiota [Title/Abstract])) OR (Microbiotas [Title/Abstract])) OR (Microbial Community [Title/Abstract])) OR (Microbial Communities [Title/Abstract])) OR (Microbiome [Title/Abstract])) OR (Microbiomes [Title/Abstract])). Studies were limited to those published from inception to April 1, 2022. In addition, we restricted the search to papers written without language restrictions.

The GMrepo (data repository for Gut Microbiota) database is a curated database of human gut metagenomes, which can provide data resources with high reusability and accessibility (35). It includes 253 projects concerning 92 phenotypes. The GMrepo database adopts a two-step quality control process to ensure data quality. Amplicon sequencing samples/runs with <20,000 reads or only a single taxon were marked as "failed QC (QC status = 0)." We searched project data from the GMrepo database (https://gmrepo.humangut.info), and the search condition was set as follows: (i) the phenotype was "autism spectrum disorder"; (ii) healthy controls were mandatory; (iii) studies with fecal microbiota analyzed using 16SrRNA sequencing; and (iv) the percentage of failed runs was set to less than 90% to eliminate the projects whose percentage of failed runs exceeds 90%.

**Data set collection.** Raw sequence data and metadata were retrieved from the NCBI Short Read Archive (SRA), European Nucleotide Archive (ENA), GMrepo database, or article's contents. Next, we excluded samples if the metadata were not available. Study information from the selected research included demographic information and methodologies, such as the recruiting area, sample size, mean age, diagnostic methods of ASD, PCR primer, sequencing platform, hypervariable region sequence, and whether or not the raw data available were recorded. The samples were grouped before downstream analyses.

**Processing of 16SrRNA gene sequences.** All downloaded raw sequencing reads of 16SrRNA gene sequences were processed using USEARCH (36). When the sequences provided by the study were paired-end sequence files, the fastq_mergepairs command was used to join the reads. Fastq_minmergelen 250 command was used to filter reads less than 250 bp, and the resulting merged reads were filtered to exclude low-quality reads using a maximum false rate of 5% as the evaluation standard. Finally, 83.5% of the original reads were retained. When building zOTU, the shortest length of reads was set to obtain the absolute abundance of zOTU, and a data normalization technique was used, namely, total sum scaling, to obtain the relative abundance of zOTU for downstream analysis. Samples with less than 6,000 valid tags were deleted. Meanwhile, when calculating alpha and beta diversity, the sequencing depth was set to 6,000. At least eight exact duplicates sequences were screened as representative sequences. zOTU was obtained by denoising with unoise3 algorithm. After identifying and removing chimeric sequences, these zOTUs were classified to the deepest taxonomic level that had 80% support using the Naive Bayesian classifier trained on the RDP taxonomy outline (version 14) (37). A total of 51,204,546 valid tags were obtained from 1,019 samples, with an average of 50,250 for each sample.

**Microbial community profiling.** The Shannon diversity index, observed OTUs, and Pielou's evenness index were used to compare the differences in alpha diversity between ASD patients and healthy controls in different projects. Stata version 16.0 (College Station, TX) was used for meta-analysis. For continuous variables, comparisons between the two groups assessed the weighted standardized mean difference (SMD) and 95% confidence intervals (CIs), and Hedges' g was selected as the final effect size. $P$-value < 0.05 was characterized as significant difference. Cochrane's Q test and $I^2$ statistics were used to assess the extent of heterogeneity among studies. A high degree of heterogeneity was expected; thus, pooled estimates was obtained through random-effect models. Beta diversity (between-sample) was assessed on the basis of Bray–Curtis dissimilarity and unweighted uniFrac distances and visualized by principal coordinate analysis (PCoA). Moreover, analysis of similarities (ANOSIM) was conducted on the basis of Bray–Curtis dissimilarity to test whether the difference between the two groups was significantly more significant than that within groups. We used the permutational multivariate analysis of variance (PERMANOVA) to assess microbiota differences between patients with ASD and healthy controls.

Next, the LEfSe method using the Kruskal–Wallis test was performed to identify the microbiota biomarkers. A linear discriminant analysis (LDA) score ($\log_{10}$) of 2.0 was used as the cutoff (38). The genera that were significantly different in relative abundance between the two groups were determined using the Wilcoxon rank-sum test. The results were considered significant at $P$ values of <0.05. LEfSe and Wilcoxon rank sum test were used to obtain the dominant genera of ASD and control group in different subgroups, and the intersection was included. Then, the union of dominant genera of ASD and control group in different subgroups was investigated. After removing the dominant genera that repeatedly existed in ASD and control groups, the remaining genera were used as microbiota biomarkers for subsequent analysis. The random forest model was used to determine whether a composite microbial biomarker could discriminate ASD versus controls using the "RandomForest" package in R (ver. 4.0.3, R Foundation for Statistical Computing). The relative importance of each genus in the predictive model was evaluated using the mean decreasing accuracy and Gini coefficient. If more genera are screened, the top 12 genera of importance were retained for

subsequent analysis. Finally, subject operating characteristic (ROC) curve analysis was used to evaluate the clinical diagnostic ability of microbial biomarkers. Furthermore, 70% data and 30% data were randomly selected as the training set and verification set, respectively.

## SUPPLEMENTAL MATERIAL

Supplemental material is available online only.

**SUPPLEMENTAL FILE 1**, PDF file, 0.8 MB.

## ACKNOWLEDGMENTS

All authors have made substantive contributions to this manuscript. Manuscript writing, sequencing analyses and management: Y.X., Y.W., J.X., Y.S.; project supervision and manuscript revision: Y.W., Z.X., and B.L.; study design, project management, financial support, and manuscript revision: B.L. and Z.X. None of the authors have financial disclosures to report. The authors have no conflicts of interest to declare.

Raw sequence data and metadata were retrieved from the NCBI SRA, ENA, and GMrepo databases. We thank the projects for the availability for data in this study. This study was supported by the National Key Research and Development Program of China (2018YFC2002000).

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
