## [Reviewer comments · Microbiology Spectrum]

Microbiology Spectrum

Leveraging Existing 16SrRNA Microbial Data to Define a Composite Biomarker for Autism Spectrum Disorder

YuShuang Xu, YiHua Wang, JinShuang Xu, Yu Song, Bingqiang Liu, and ZhiFan Xiong

Corresponding Author(s): YuShuang Xu, Liyuan Hospital, Tongji Medical College, Huazhong University of Science and Technology, WuHan, China

Review Timeline:

Submission Date:	January 26, 2022
Editorial Decision:	March 25, 2022
Revision Received:	May 25, 2022
Accepted:	June 3, 2022

Editor: Kristen DeAngelis

Reviewer(s): The reviewers have opted to remain anonymous.

Transaction Report:

DOI: <https://doi.org/10.1128/spectrum.00331-22>

March 25, 2022

Dr. YuShuang Xu
Liyuan Hospital, Tongji Medical College, Huazhong University of Science and Technology, WuHan, China
Gastroenterology
YanHu Avenue
Wuhan, Hubei 430077
China

Re: Spectrum00331-22 (Leveraging Existing 16SrRNA Microbial Data to Define a Composite Biomarker for Autism Spectrum Disorder)

Dear Dr. YuShuang Xu:

Thank you for submitting your manuscript to Microbiology Spectrum. This research topic is of interest to the readership of ASM Spectrum, but both reviewers had substantial concerns with the nature of the datasets, how the data were analyzed, and the analytic methods used to support the conclusions. The reviewers' comments would need to be addressed, and the manuscript revised for clarity, before we could consider publication.

Link Not Available

Sincerely,

Kristen DeAngelis

Journals Department
Reviewer comments:

Reviewer #1 (Comments for the Author):

The authors collected raw data from 12 studies of gut/oral microbiota studies associated with ASD, and reanalyzed them. The results showed significant differences in fecal and oral microbiota composition between ASD and healthy controls, but only fecal microbial markers (18 genera) are of great significance in diagnosing ASD diseases. Though interesting, there are several problems affecting the quality of this manuscript. So it must be improved before

publication.

Major concerns:

1. More details should be provided in Processing of 16SrRNA gene sequences. Since each study used different sequencing regions, like V2-V3, V3-V4, V4-V5, V4, and even unknown, how can you combine them together for OTU construction? In addition, do you analyze fecal samples and oral samples separately, or combine them into one dataset to construct OTU? Did you normalize the reads number in OTU construction?
2. You mentioned the study of Yap CX, which concluded dietary preferences caused the difference in gut microbiota. So can you exclude the affection of dietary difference in your study, or adjust this confounding factor?

Minor concerns:

- 1 I think Extra tree (Line 227-228) is different with Random Forest (Line 157), please clarify.
2. Line 142-143, it's difficult to understand.
3. Many typographical errors were found in the manuscript, like Line 86-87, Line 133-135, etc. Please check it carefully and find a native English speaker for language proofing.
4. Line 155, Wilcoxon test. Do you perform Wilcoxon rank-sum test or signed-rank test? How do you define significant difference, using P-value or FDR?
5. Line 209-226, it's hard to follow. How do you define overlapped genera? For example, in different variable regions, the obtained 33 genera is a simple overlap between Table S4 and 5? or must be found in more than three subgroups? I suggest you add a supplementary figure to illustrate this paragraph more clearly.
6. Line 226, Figure 3d? There's no this panel.
7. 'five studies was shown in Figures 2e&f. '? It should be five different variable regions.
8. A Limitation part is needed in the last of Discussion.

Reviewer #3 (Comments for the Author):

This study identified gut and oral microbiome as a biomarker for ASD using 10 and 2 existing datasets, respectively. An increasing number of 16S rRNA gene sequencing datasets on the gut microbiome have encouraged researchers to identify biomarkers for ASD phenotype, but complex ASD symptoms and different sequencing assays applied have been a great challenge. Here are major and minor comments below.

Major comments:

1. LefSe and Wilcoxon tests were utilized to find the differential taxa between ASD and healthy groups, and non-differential genus and differential genus were noted as "0" and "1", respectively, in Tables S4-S6. This approach does not include whether the individual genus is either more or less abundant in ASD samples. For example, some studies reported *Desulfovibrio* are more abundant in ASD group, but other studies found they are less abundant in ASD group. The results in this manuscript could be biased by this inconsistent taxon that is different between ASD and control groups but inconsistent in their abundances across studies.
2. Data set collection (L123-129): Authors ended up 10 gut microbiome datasets out of 253 projects after rigorous screening procedures. According to Table 1, however, 50% of datasets (5 out of 10) were not available with ages, which is not a small concern. The gut microbiome substantially changes especially in early ages (Yatsunenko et al. 2012 Nature). Information on two datasets (S9 and S10) are minimal as they were not published previously. Authors brought up a good amount of discussion on diet in lines 341-350, but the current study did not have dietary components included in the analysis. What about other parameters such as gender, GI issues, and ASD severity? These are significant factors that drive the gut microbiome in different ways, but they were not considered in the current analysis, either. Without comprehensive metadata included, biomarkers claimed in this study could be limited to differentiate ASD group against control group correctly.
3. Training set vs. verification sets: Authors need to explain how training and verification sets were prepared for both gut and oral microbiome analyses. A total of 18 genera were identified as biomarkers together. However, 18 genera are too large to be universally applied as biomarkers. It is suggested to test a subset of 18 genera could still be efficient to differentiate the groups.
4. Oral microbiome set: In this study, only two datasets were obtained for the analysis, and a literature review on the oral microbiome is very limited in Introduction (lines 83-84). Two datasets are too small to identify and claim any biomarker for ASD. Authors may consider focusing only gut microbiome to identify its biomarker in this manuscript.

Minor comments:

1. Authors used the LDA score (log₁₀) of 2.0 as the cutoff, but the score of 2.0 is relatively insignificant to rely on. There need references to support the decision on this cutoff.
2. Authors need to explain which criteria were used to claim significance on Wilcoxon test. No explanation (such as p-values and multiple-hypothesis correction) is available in Methods.
3. In lines 210-211, 215-217, and 234-235, different groups of genera were recognized as important taxa, but it is not clear how these taxa were determined.
4. The materials/methods related to Figure 2e-h are very limited and difficult to understand.
5. Lines 121: Not clear "4) the percent of failed runs was less than 90%."
6. Line 157: Not clear "... and token the intersection"
7. Line 226: Figure 3d should be Figure 2d.
8. Line 227: It is not clear which genera were indicated in the sentence "...we used these differential genera..."
9. Line 228: What are 5 different bacteria?
10. PCoA based on Bray-Curtis dissimilarity (Figure 2b-d): It is not easy to differentiate colors between groups in PCoA.
11. References are missing in lines 73-75, 75-77, 305-308, and 309-311.

Staff Comments:

Preparing Revision Guidelines

Please return the manuscript within 60 days; if you cannot complete the modification within this time period, please contact me. If you do not wish to modify the manuscript and prefer to submit it to another journal, please notify me of your decision immediately so that the manuscript may be formally withdrawn from consideration by Microbiology Spectrum.

Dear Editors and Reviewers:

Thank you for your letter and the reviewers' comments concerning our manuscript entitled "Leveraging Existing 16SrRNA Microbial Data to Define a Composite Biomarker for Autism Spectrum Disorder" (ID: Spectrum00331-22). Those comments are all valuable and very helpful for revising and improving our paper and the essential guiding significance to our research. We have studied the comments carefully and have made a correction which we hope meets with approval. Revised portions are marked in red in the paper. The major corrections in the paper and the responses to the reviewer's comments are as follows:

Responds to the reviewer's comments:

Reviewer #1:

Major concerns:

1. Response to comment: (More details should be provided in Processing of 16SrRNA gene sequences. Since each study used different sequencing regions, like V2-V3, V3-V4, V4-V5, V4, and even unknown, how can you combine them together for OTU construction? In addition, do you analyze fecal samples and oral samples separately, or combine them into one dataset to construct OTU? Did you normalize the reads number in OTU construction?)

Response: Thank you for your reminder and we have re-written the part of data processing process according to the Reviewer's suggestion. In the previous version, we constructed OTU separately by separating fecal and oral samples from the data set, but considering the reviewers' suggestions that only two oral research projects were included, we removed the content of oral samples. In fecal samples, we gather the original data of multiple studies and combine them together to construct zOTU. As proposed by the reviewer, different research projects involve different sequencing platforms and sequencing regions, which may affect the final result of clustering. However, we have consulted several similar literatures and found that they are data processing processes with similar methods, suggesting the feasibility of this method (Reference: PMID: 32291348) . In addition, we re-searched the database and two recent studies were included. Because the original S9 data sample size is small and

the metadata information is incomplete, it is eliminated. Only the Illumina sequencing platform results are retained to minimize the heterogeneity of the project. The original S8 data was deleted because the Genome Sequencer FLX-Titanium was used. The following is the revised part of the data processing process: All downloaded raw sequencing reads of 16SrRNA gene sequences were processed using USEARCH. When the sequences provided by the study were paired-end sequence files, the `fastq_mergepairs` command was used to join the reads. `Fastq_minmergelen 250` command was used to filter reads less than 250 bp, and the resulting merged reads were filtered to exclude low-quality reads using a maximum false rate of 5% as the evaluation standard. Finally, 83.5% of the original reads were retained. When building zOTU, the shortest length of reads was set to obtain the absolute abundance of zOTU, and a data normalization technique was used, namely, total sum scaling, to obtain the relative abundance of zOTU for downstream analysis. Samples with less than 6000 valid tags were deleted. Meanwhile, when calculating alpha and beta diversity, the sequencing depth was set to 6000. At least eight exact duplicates sequences were screened as representative sequences. zOTU was obtained by denoising with `unoise3` algorithm. After identifying and removing chimeric sequences, these zOTUs were classified to the deepest taxonomic level that had 80% support using the Naive Bayesian classifier trained on the RDP taxonomy outline (version 14). A total of 51,204,546 valid tags were obtained from 1019 samples, with an average of 50,250 for each sample.

2. Response to comment: (You mentioned the study of Yap CX, which concluded dietary preferences caused the difference in gut microbiota. So can you exclude the affection of dietary difference in your study, or adjust this confounding factor?)

Response: As the reviewer proposed that we can't exclude the affection of dietary differences or adjust this confounding factor. Previous studies have shown that dietary patterns and composition affect gut microbiota composition (Reference: PMID: 31541197). Lack of access to dietary information is an important limitation of this study and a common problem in the current study of gut microbiota. We added this to the part of Strength and limitation, and suggested adding and standardizing the factors

affecting the gut microbiota. The following are the revised part of Strength and limitation: This study can identify some studies and pool their results, with a relatively large number of participants. It is also an update and extension of previous similar studies. Nevertheless, this study has several limitations. First, the metadata information of existing research is incomplete. The composition of individual feces varies greatly, which is affected by many confounding factors, such as region, diet, exercise, gender, age, body mass index, and specimen collection method. Therefore, we hope to integrate other possible confounding factors in future research.

Minor concerns:

1. Response to comment: (I think Extra tree (Line 227-228) is different with Random Forest (Line 157), please clarify.)

Response: It is true as the Reviewer proposed that extremely randomized trees is different with Random Forest. We have re-written this part and following is revised contents: We used the dominant genera identified in different subgroups to build prediction models based on the random forest model.

2. Response to comment: (Line 142-143, it's difficult to understand.)

Response: Thank you for your reminder. The raw data were subjected to quality control in data analysis, and the clean tags were obtained after removing the repeated sequences. After removing the chimera, the final valid tags for the subsequent zOTU division were obtained. We made statistics on the number of the valid tags. We have re-written this part, and the following is the revised contents: A total of 51,204,546 valid tags were obtained from 1019 samples, with an average of 50,250 for each sample.

3. Response to comment: (Many typographical errors were found in the manuscript, like Line 86-87, Line 133-135, etc. Please check it carefully and find a native English speaker for language proofing.)

Response: We are very sorry for our incorrect writing and the listed problems have been modified. "The differences in species richness, diversity, composition, and structure of the gut microbiota between ASD patients and controls using 16S sequencing have been repeatedly reported, and they have received intense attention;

however, a consensus among studies is rarely reported.” “When the sequences provided by the study were paired-end sequence files, the fastq_mergepairs command was used to join the reads. Fastq_minmergelen 250 command was used to filter reads less than 250 bp, and the resulting merged reads were filtered to exclude low-quality reads using a maximum false rate of 5% as the evaluation standard.” In addition, we revised the manuscript sentence by sentence and the language of our manuscript has been polished by a professional editing company.

4. Response to comment: (Line 155, Wilcoxon test. Do you perform Wilcoxon rank-sum test or signed-rank test? How do you define significant difference, using P-value or FDR?)

Response: We are very sorry for not describing it clearly. The genera that were significantly different in relative abundance between the two groups were determined using the Wilcoxon rank-sum test. The results were considered significant at P values of <0.05 .

5. Response to comment: (Line 209-226, it's hard to follow. How do you define overlapped genera? For example, in different variable regions, the obtained 33 genera is a simple overlap between Table S4 and 5? or must be found in more than three subgroups? I suggest you add a supplementary figure to illustrate this paragraph more clearly.)

Response: Thank you for your reminder. We have carefully described the process of selecting the genera in the section of Microbial community profiling. The following are the added parts: Next, the LEfSe method using the Kruskal–Wallis test was performed to identify the microbiota biomarkers. A linear discriminant analysis (LDA) score (\log_{10}) of 2.0 was used as the cutoff. The genera that were significantly different in relative abundance between the two groups were determined using the Wilcoxon rank-sum test. The results were considered significant at P values of <0.05 . LEfSe and Wilcoxon rank sum test were used to obtain the dominant genera of ASD and control group in different subgroups, and the intersection was included. Then, the union of dominant genera of ASD and control group in different subgroups was investigated. After removing the dominant genera that repeatedly existed in ASD

and control groups, the remaining genera were used as microbiota biomarkers for subsequent analysis.

6. Response to comment: (Line 226, Figure 3d? There's no this panel.)

Response: We are very sorry for our incorrect writing. We have checked and re-uploaded all the pictures.

7. Response to comment: ('five studies was shown in Figures 2e&f. '? It should be five different variable regions.)

Response: We are very sorry for our incorrect writing and the listed problems have been modified.

8. Response to comment: (A Limitation part is needed in the last of Discussion.)

Response: Thank you for your reminder. The following are the added part of strength and limitation: This study can identify some studies and pool their results, with a relatively large number of participants. It is also an update and extension of previous similar studies. Nevertheless, this study has several limitations. First, the metadata information of existing research is incomplete. The composition of individual feces varies greatly, which is affected by many confounding factors, such as region, diet, exercise, gender, age, body mass index, and specimen collection method. Therefore, we hope to integrate other possible confounding factors in future research. Second, the projects included in this study involve different sequencing platforms and variable regions targeted. We tried to eliminate the batch effect by using the combat function and removeBatchEffect limma function, but it was not successful. Large batch effects can still be observed among different projects. Therefore, the method of eliminating the batch effect must be developed and utilized more. Third, whether intestinal disorders precede the development of ASD or whether restrictive and repetitive behaviors such as diet in ASD drive this disorder remain unclear. The dynamic changes in the relative abundance of individual features in time sequence may be used to identify the driving pathogenic factors in the development of ASD. Large-scale human microbiota research must collect samples from multiple time points to maximize the detection of minor effects in microbial host interactions.

Reviewer #3:

Major comments:

1. Response to comment: (LefSe and Wilcoxon tests were utilized to find the differential taxa between ASD and healthy groups, and non-differential genus and differential genus were noted as "0" and "1", respectively, in Tables S4-S6. This approach does not include whether the individual genus is either more or less abundant in ASD samples. For example, some studies reported *Desulfovibrio* are more abundant in ASD group, but other studies found they are less abundant in ASD group. The results in this manuscript could be biased by this inconsistent taxon that is different between ASD and control groups but inconsistent in their abundances across studies.)

Response: As proposed by the Reviewer, we did not distinguish the dominant bacteria in ASD group and control group, and also ignored the differences in the results of some genera in different studies. We have carefully described the process of selecting the genera in the section of Microbial community profiling. The following are the added parts: Next, the LEfSe method using the Kruskal–Wallis test was performed to identify the microbiota biomarkers. A linear discriminant analysis (LDA) score (log₁₀) of 2.0 was used as the cutoff. The genera that were significantly different in relative abundance between the two groups were determined using the Wilcoxon rank-sum test. The results were considered significant at P values of <0.05. LEfSe and Wilcoxon rank sum test were used to obtain the dominant genera of ASD and control group in different subgroups, and the intersection was included. Then, the union of dominant genera of ASD and control group in different subgroups was investigated. After removing the dominant genera that repeatedly existed in ASD and control groups, the remaining genera were used as microbiota biomarkers for subsequent analysis.

2. Response to comment: (Data set collection (L123-129): Authors ended up 10 gut microbiome datasets out of 253 projects after rigorous screening procedures. According to Table 1, however, 50% of datasets (5 out of 10) were not available with ages, which is not a small concern. The gut microbiome substantially changes especially in early ages (Yatsunenko et al. 2012 Nature). Information on two datasets

(S9 and S10) are minimal as they were not published previously. Authors brought up a good amount of discussion on diet in lines 341-350, but the current study did not have dietary components included in the analysis. What about other parameters such as gender, GI issues, and ASD severity? These are significant factors that drive the gut microbiome in different ways, but they were not considered in the current analysis, either. Without comprehensive metadata included, biomarkers claimed in this study could be limited to differentiate ASD group against control group correctly.)

Response: We are sorry that there was an error in the search strategy in the previous version. Therefore, we re-searched within the PubMed database. The search strategy and search terms were constructed as follows: (((((Autism Spectrum Disorder [MeSH Terms]) OR (Autism Spectrum Disorder [Title/Abstract])) OR (Autistic Spectrum Disorder [Title/Abstract])) OR (Autism [Title/Abstract])) OR (Autistic Disorder [Title/Abstract])) AND ((((((Microbiota [MeSH Terms]) OR (Microbiota [Title/Abstract])) OR (Microbiotas [Title/Abstract])) OR (Microbial Community [Title/Abstract])) OR (Microbial Communities [Title/Abstract])) OR (Microbiome [Title/Abstract])) OR (Microbiomes [Title/Abstract])). Studies were limited to those published from inception to 1st April 2022. In addition, we restricted the search to papers written without language restrictions. The flowchart of study selection is shown in Figure 1. The systematic searches produced 792 records from the PubMed database and ten projects from the GMrepo database. After preliminary screening of titles and abstracts, 51 records and one project were identified for full-text review. Because the original S9 data sample size is small and the metadata information is incomplete, it is eliminated. We only retained the Illumina sequencing platform results to minimize project heterogeneity, therefore, the original S8 data was deleted because the Genome Sequencer FLX-Titanium was used. The original S10 retrieved the published literature from Ding *et al.*, which is the modified S9. Ultimately, the 16SrRNA gene sequencing data from 10 microbiota studies met the criteria for further analysis. The ten data sets were labeled as S1-S10. This combined data set consisted of 1019 participants (569 ASD and 450 healthy controls), and the study sizes varied from 12 to 286 subjects. The sequencing platforms of all studies are Illumina. The

sequencing fragments include V3-4, V4 and V4-5, and the research countries include China, Ecuador, Italy and Korean. As the Reviewer suggested that the gut microbiome substantially changes especially in early ages. In 10 data sets, except S1 data set, the other 9 data sets have average age information. The results are shown in Table 1. The subjects of study table item S1 are minors. We could not obtain the age, gender, gastrointestinal problem and autism severity of each sample, so we could not judge the impact of these factors on the results. This is also a major defect in the current study of gut microbiota. We added it to limitations, and the following is the added part of Strength and limitation: This study can identify some studies and pool their results, with a relatively large number of participants. It is also an update and extension of previous similar studies. Nevertheless, this study has several limitations. First, the metadata information of existing research is incomplete. The composition of individual feces varies greatly, which is affected by many confounding factors, such as region, diet, exercise, gender, age, body mass index, and specimen collection method. Therefore, we hope to integrate other possible confounding factors in future research.

3. Response to comment: (Training set vs. verification sets: Authors need to explain how training and verification sets were prepared for both gut and oral microbiome analyses. A total of 18 genera were identified as biomarkers together. However, 18 genera are too large to be universally applied as biomarkers. It is suggested to test a subset of 18 genera could still be efficient to differentiate the groups.)

Response: Thank you for your comments. We use the the mean decreasing accuracy and the Gini coefficient to evaluate the relative importance of each genus in the predictive model. If more species of genera are screened, then the 12 genera were retained for subsequent analysis. Finally, subject operating characteristic (ROC) curve analysis was used to evaluate the clinical diagnostic ability of microbial biomarkers. Furthermore, 70% data and 30% data were randomly selected as the training set and verification set, respectively. Next, we used the dominant genera identified in different subgroups to build prediction models based on the random forest model. The top 12 important genera in the ASD and healthy control groups determined by

subgroup analysis of different studies can distinguish the two groups, and the area under the receiver operating characteristic curve (AUC) of the training set and verification set was 0.688 and 0.706, respectively (Figure 4a). The eight genera in the ASD and healthy control groups determined by subgroup analysis of different variable regions also can distinguish the two groups, and the AUC of the training set and verification set was 0.725 and 0.658, respectively (Figure 4b). By contrast, the 12 genera in the ASD and healthy control groups determined by subgroup analysis of different sequencing platforms could distinguish ASD from control, and the AUC of the training set and verification set was 0.757 and 0.761, respectively (Figure 4c).

4. Response to comment: (Oral microbiome set: In this study, only two datasets were obtained for the analysis, and a literature review on the oral microbiome is very limited in Introduction (lines 83-84). Two datasets are too small to identify and claim any biomarker for ASD. Authors may consider focusing only gut microbiome to identify its biomarker in this manuscript.)

Response: As suggested by the reviewer that the literature review on the oral microbiota is very limited. Considering that only two oral research projects were included in this analysis, we removed the contents of the oral sample.

Minor comments:

1. Response to comment: (Authors used the LDA score (log10) of 2.0 as the cutoff, but the score of 2.0 is relatively insignificant to rely on. There need references to support the decision on this cutoff.)

Response: Thank you for your reminder. We have added references (Reference: PMID: 25882912).

2. Response to comment: (Authors need to explain which criteria were used to claim significance on Wilcoxon test. No explanation (such as p-values and multiple-hypothesis correction) is available in Methods.)

Response: We are very sorry for not describing it clearly. The genera that were significantly different in relative abundance between the two groups were determined using the Wilcoxon rank-sum test. The results were considered significant at *P* values

of <0.05 .

3. Response to comment: (In lines 210-211, 215-217, and 234-235, different groups of genera were recognized as important taxa, but it is not clear how these taxa were determined.)

Response: Thank you for your reminder. We have carefully described the process of selecting the genera in the section of Microbial community profiling. The following are the added parts: Next, the LEfSe method using the Kruskal–Wallis test was performed to identify the microbiota biomarkers. A linear discriminant analysis (LDA) score (\log_{10}) of 2.0 was used as the cutoff. The genera that were significantly different in relative abundance between the two groups were determined using the Wilcoxon rank-sum test. The results were considered significant at P values of < 0.05 . LEfSe and Wilcoxon rank sum test were used to obtain the dominant genera of ASD and control group in different subgroups, and the intersection was included. Then, the union of dominant genera of ASD and control group in different subgroups was investigated. After removing the dominant genera that repeatedly existed in ASD and control groups, the remaining genera were used as microbiota biomarkers for subsequent analysis.

4. Response to comment: (The materials/methods related to Figure 2e-h are very limited and difficult to understand.)

Response: We revised and re-uploaded the figures. Figure 3e-h refer to the overlapping feature of dominant genera in the ASD group between studies S1–10 using LEfSe algorithm (e) and Wilcoxon rank-sum test (f). And the overlapping feature of dominant genera in the control group between studies S1–10 using LEfSe algorithm (g) and Wilcoxon rank-sum test (h).

5. Response to comment: (Lines 121: Not clear "4) the percent of failed runs was less than 90%.")

Response: We are very sorry for not describing it clearly. Referring to the literature of Li *et al.*, GMrepo is a database of curated and consistently annotated human gut metagenomes. The database adopts a two-step quality control process to ensure data quality. First, amplicon sequencing samples/runs with $<20\ 000$ reads were removed

from subsequent analysis and were marked as ‘failed QC (QC status = 0)’ in GMrepo. The second step of quality control is for both amplicon sequences and metagenomic sequences. After taxonomy assignment, samples/runs containing only a single taxon, i.e., a species or a genus accounted for more than or equal to 99.99% of the total abundance, will also be marked as ‘failed QC (QC status = 0)’. In order to eliminate the projects whose percentage of failed runs exceeds 90%, we set the filter criteria to the percent of failed runs was less than 90%. Following is the revised parts: The GMrepo (data repository for Gut Microbiota) database is a curated database of human gut metagenomes, which can provide data resources with high reusability and accessibility. It includes 253 projects concerning 92 phenotypes. The GMrepo database adopts a two-step quality control process to ensure data quality. Amplicon sequencing samples/runs with <20 000 reads or only a single taxon were marked as “failed QC (QC status = 0).” We searched project data from the GMrepo database (<https://gmrepo.humangut.info>), and the search condition was set as follows:(1) the phenotype was “autism spectrum disorder”; (2) healthy controls were mandatory; (3) studies with fecal microbiota analyzed using 16SrRNA sequencing; (4) the percentage of failed runs was set to less than 90% to eliminate the projects whose percentage of failed runs exceeds 90%.

6. Response to comment: (Line 157: Not clear "... and token the intersection")

Response: We are very sorry for not describing it clearly. We rewrote the selection process of dominant genera in ASD group and control group. The following is the revised part of Microbial community profiling: Next, the LEfSe method using the Kruskal–Wallis test was performed to identify the microbiota biomarkers. A linear discriminant analysis (LDA) score (log₁₀) of 2.0 was used as the cutoff. The genera that were significantly different in relative abundance between the two groups were determined using the Wilcoxon rank-sum test. The results were considered significant at *P* values of <0.05. LEfSe and Wilcoxon rank sum test were used to obtain the dominant genera of ASD and control group in different subgroups, and the intersection was included. Then, the union of dominant genera of ASD and control group in different subgroups was investigated. After removing the dominant genera

that repeatedly existed in ASD and control groups, the remaining genera were used as microbiota biomarkers for subsequent analysis.

7. Response to comment: (Line 226: Figure 3d should be Figure 2d)

Response: Response: We are very sorry for our incorrect writing and checked and re-uploaded all the figures.

8. Response to comment: (Line 227: It is not clear which genera were indicated in the sentence "...we used these differential genera...")

Response: We are very sorry for not describing it clearly. Differential genera refer to dominant bacteria in ASD and control group.

9. Response to comment: (Line 228: What are 5 different bacteria?)

Response: We are very sorry for not describing it clearly. Differential genera refer to dominant bacteria in ASD and control group.

10. Response to comment: (PCoA based on Bray-Curtis dissimilarity (Figure 2b-d): It is not easy to differentiate colors between groups in PCoA.)

Response: Thank you for your reminder. We have changed the color of the group and re-uploaded all the figures.

11. Response to comment: (References are missing in lines 73-75, 75-77, 305-308, and 309-311.)

Response: We are very sorry for our mistakes and the listed problems have been modified.

We tried our best to improve the manuscript and made some changes in the manuscript. A professional editing company has polished the language of our manuscript. These changes will not influence the content and framework of the paper. And here, we did not list the changes but marked them in red in the revised paper.

We appreciate Reviewers' warm work earnestly and hope that the correction will meet with approval. Once again, thank you very much for your comments and suggestions.

Yours sincerely,

ZhiFan Xiong

June 3, 2022

Dr. YuShuang Xu
Liyuan Hospital, Tongji Medical College, Huazhong University of Science and Technology, WuHan, China
Gastroenterology
YanHu Avenue
Wuhan, Hubei 430077
China

Re: Spectrum00331-22R1 (Leveraging Existing 16SrRNA Microbial Data to Define a Composite Biomarker for Autism Spectrum Disorder)

Dear Dr. YuShuang Xu:

Thank you for your careful attention to the reviewers comments. Your manuscript has been accepted, and I am forwarding it to the ASM Journals Department for publication. You will be notified when your proofs are ready to be viewed.

Sincerely,

Kristen DeAngelis
Editor, Microbiology Spectrum
